# INTERPRETABLE CONTINUAL LEARNING

## ABSTRACT

We present a framework for *interpretable continual learning* (ICL). We show that explanations of previously performed tasks can be used to improve performance on future tasks. ICL generates a good explanation of a finished task, then uses this to focus attention on what is important when facing a new task. The ICL idea is general and may be applied to many continual learning approaches. Here we focus on the variational continual learning framework to take advantage of its flexibility and efficacy in overcoming catastrophic forgetting. We use saliency maps to provide explanations of performed tasks and propose a new metric to assess their quality. Experiments show that ICL achieves state-of-the-art results in terms of overall continual learning performance as measured by average classification accuracy, and also in terms of its explanations, which are assessed qualitatively and quantitatively using the proposed metric.

## 1 INTRODUCTION

Continual learning, also called lifelong learning, refers to frameworks where knowledge acquired from past tasks is accumulated for use on future tasks, i.e. where learning continually proceeds in an online fashion. Data belonging to different tasks might be non i.i.d. (Nguyen et al., 2018; Ring, 1997; Schmidhuber, 2013; Schlimmer & Fisher, 1986; Sutton & Whitehead, 1993). Crucially, a continual learner must be able to learn a new task without forgetting how to perform previous tasks (Ring, 1995; Schwarz et al., 2018). Continual learning frameworks need to continually adapt to the domain shift occurring across tasks, without revisiting data from previous tasks. An appropriate balance is required between stability and adapting to new tasks and data, since excessive adaptation might lead to dramatic degradation in performance of earlier tasks, known as *catastrophic forgetting* (French, 1999; Goodfellow et al., 2014; McCloskey & Cohen, 1989; Ratcliff, 1990).

Several approaches have been introduced to address catastrophic forgetting. One approach is based on regularisation where stability is maintained by restricting the change of parameters with high influence while allowing the other parameters to vary freely (Li & Hoiem, 2016; Nguyen et al., 2018; Kirkpatrick et al., 2017; Vuorio et al., 2018; Zenke et al., 2017). Another approach divides the network architecture into reusable parts that are less prone to changes, and parts devoted to individual tasks (Fernando et al., 2017; Rusu et al., 2016b;a; Yoon et al., 2018). The framework in (Xu & Zhu, 2018) constructs the neural network architecture via designed reinforcement learning (RL) strategies. The framework in (Lee et al., 2017) bases its solution on moment matching. Another RL based framework is presented in (Kaplanis et al., 2018) where catastrophic forgetting is mitigated at multiple time scales by using RL agents with a synaptic model. Mankowitz et al. (2018) propose a framework where multiple agents jointly learn to achieve multiple goals at once in a parallel off-policy setup. The work in (Farquhar & Gal, 2018) proposes experimental evaluations of continual learning as well as a variational Bayesian loss, via which they categorise a few previous works into either prior-focused or likelihood-focused. Attention mechanisms have been developed in rather similar problems before, e.g. in (Khetarpal & Precup, 2018; Serra et al., 2018; Welleck et al., 2017; Heo et al., 2018). Other saliency metrics have been introduced in (Dabkowski & Gal, 2017; Adebayo et al., 2018). To the best of our knowledge, our framework is the first piece of work to pursue a comprehensive interpretability approach in the continual learning setting.

Our work is based on the idea of imitating some aspects of how humans learn continually. We suggest that humans are quite successful in achieving goals and performing tasks sequentially, partly because we manage to understand and explain to ourselves certain aspects of the tasks we have already accomplished. This provides a contribution to our performance on similar tasks in the future.

When given a task, we typically not only accomplish it, but also often (perhaps unconsciously) gain a useful interpretable concept. For instance, when a child tries to grasp an object, the progress in the child's skill is accompanied by improvements in similar tasks, e.g. grasping other objects which the child has never seen before. The development of the child's cognitive abilities shows an understanding of the concepts of gravity, geometric characteristics of the object, etc (Schroeder, 2014). We suggest that this development may be established by the consolidation of interpretable information from past experience. With this motivation, here we consider and analyse the impact of interpretable methods on continual learning. While there can be a tradeoff between performance and interpretability when considering just one task, here we show that interpretability can help performance when a learner faces consecutive tasks over time, as in continual learning.

We propose a continual learning framework where the training phase of each task is followed by an explanation stage, which provides insights to be utilised along with the established training platform in learning the subsequent tasks. We focus on image classification tasks where training of the first task proceeds normally, before providing a saliency map for each test data point (image). By highlighting the most relevant areas of the test image w.r.t. the classification prediction, an understanding of the individual decisions taken by the classifier is attained. Afterwards, we compute a saliency map representing a summary (or average) of the saliency maps of test images per class, i.e. a summary of the classifier's decisions on the test data. This represents a summarised belief of the relevant areas of the input for each class. After completing the current task, we can use the explanation, depicted by the average saliency map, to achieve two goals:

i Assessing how good the developed continual learning framework is at eradicating catastrophic forgetting. This evaluation is typically performed by measuring the difference in the classification performance on a certain task when it is the most recently encountered vs. when other tasks subsequently followed. Here we propose a new measure by adding a test which compares the saliency map resulting from testing a task right after finishing the corresponding training phase, with the saliency map of the same test on the same task after having other tasks subsequently trained by our continual learner. Degradation of the provided explanations provides a measure of the level of catastrophic forgetting induced in the continual learning framework.
ii Providing interpretable attention information for subsequent tasks by involving the obtained saliency maps of the current task in the optimisation for the future tasks.

To achieve the first goal, we need a metric to assess the quality of the extracted saliency maps. We propose a new metric for evaluating saliency maps resulting from the classification decisions on test data. Measuring the quality of a saliency map is not a straightforward task. Saliency maps typically aim at explaining the classification decision taken by a classifier. In other words, a saliency map seeks the subset of features that are the most influential in the resulting prediction of the classifier. As such, the explanation provided by a saliency map comes out in the form of a summary of the significant parts (features) of the input data, from the classifier's point of view. We propose a metric to assess the quality of an explanation resulting from a saliency map.

To achieve the second goal, we need to involve the saliency maps in the learning procedure, not just during the test phase. We propose an attention mechanism that exploits the feature relevance values learnt in the latest task to focus the attention of the new learner on what is believed to be the most important parts of the input as per the latest task, which (w.r.t. continual learning) is assumed to be similar. Thus, an explanation of a task evolving over time is utilised to help the emerging tasks.

Note that there is a difference in the nature of the two proposed goals described above. The optimisation needed to achieve the second goal does not guarantee that the first goal is automatically achieved, since the first goal addresses explanations (saliency maps) related to the same task at different time steps, whereas the second goal is concerned with exchanging interpretable information among different tasks. Hence, the assessment involved in the first goal is still needed regardless of the level of perfection of the second goal.

In this work we perform experiments for the classification case illustrated above, but the same paradigm could be applied in future work to tasks other than classification, where the personalisation or notion of the explanation will need to be adapted to the nature of the task, i.e. saliency maps should be replaced with something more suitable to the task.

We highlight the following contributions of our framework: 1) the interpretable continual learning (ICL) framework where, in addition to the ordinary understanding benefits of interpretable frame-

works, explanations of the finished tasks are used to enhance the attention of the learner during the future tasks. Although we focus on performing within the variational continual learning framework (Nguyen et al., 2018), our proposed methodology is flexible and can be deployed with other continual learning frameworks, and also with other saliency detection methods. To the best of our knowledge, ICL represents a novel direction in the continual learning literature that focuses on interpretability; 2) introducing a new metric for saliency maps that aims at robustly assessing their quality without significant engineering required, i.e. no need to construct bounding rectangles of the relevant zones or similar; 3) learning from past experiences using an attention mechanism based on explanations (saliency maps) from the latest task; 4) Our quantitative and qualitative state-of-the-art results in four experiments on three datasets demonstrate the efficacy of the proposed framework.

## 2 VARIATIONAL CONTINUAL LEARNING (VCL)

In this paper, we use Variational Continual Learning (VCL, Nguyen et al., 2018) as the underlying continual learning framework. This is a variational Bayesian framework where the posterior of the model parameters $\boldsymbol{\theta}$ is learned and updated continually from a sequence of $T$ datasets, $\{\boldsymbol{x}_t^{(n)}, \boldsymbol{y}_t^{(n)}\}_{n=1}^{N_t}$, where $t = 1, 2, \ldots, T$ and $N_t$ is the size of the $t$-th dataset. More specifically, denote by $p(\boldsymbol{y}|\boldsymbol{\theta}, \boldsymbol{x})$ the probability distribution returned by a discriminative classifier with input $\boldsymbol{x}$, output $\boldsymbol{y}$ and parameters $\boldsymbol{\theta}$. For $\mathcal{D}_t = \{\boldsymbol{y}_t^{(n)}\}_{n=1}^{N_t}$, we approximate the intractable posterior $p(\boldsymbol{\theta}|\mathcal{D}_{1:t})$ after observing the first $t$ datasets via a tractable variational distribution $q_t$ by:[1]

$$q_t(\boldsymbol{\theta}) \approx \frac{1}{Z_t} q_{t-1}(\boldsymbol{\theta}) \, p(\mathcal{D}_t|\boldsymbol{\theta}), \tag{1}$$

where $q_0$ is the prior $p$, $p(\mathcal{D}_t|\boldsymbol{\theta}) = \prod_{n=1}^{N_t} p(\boldsymbol{y}_t^{(n)}|\boldsymbol{\theta}, \boldsymbol{x}_t^{(n)})$, and $Z_t$ is the normalizing constant. This framework allows the approximate posterior $q_t(\boldsymbol{\theta})$ to be updated continuously from the previous approximate posterior $q_{t-1}(\boldsymbol{\theta})$ in an online fashion. In VCL, the approximation in Equation 1 is performed by minimizing the following KL divergence over a family $\mathcal{Q}$ of tractable distributions:

$$q_t(\boldsymbol{\theta}) = \arg\min_{q \in \mathcal{Q}} \text{KL}\Big(q(\boldsymbol{\theta}) \,\|\, \frac{1}{Z_t} q_{t-1}(\boldsymbol{\theta}) \, p(\mathcal{D}_t|\boldsymbol{\theta})\Big). \tag{2}$$

This framework can be enhanced to further mitigate the catastrophic forgetting problem by including a coreset, a small representative set of data from previously observed tasks that can serve as an episodic memory and can be revisited before a decision needs to be made.

VCL and coreset VCL are current state-of-the-art methods for continual learning that can reduce catastrophic forgetting and outperform other methods such as Elastic Weight Consolidation (EWC, Kirkpatrick et al., 2017) and Synaptic Intelligence (SI, Zenke et al., 2017) on a variety of continual learning benchmarks such as Permuted MNIST, Split MNIST and Split notMNIST. For that reason, we choose VCL as the underlying continual learning framework in our paper.

## 3 EXPLANATIONS OF CLASSIFICATION DECISIONS

Saliency maps are methods used to detect the relevance of each part of a given image for a specific class label, according to the model. For each given test point $\boldsymbol{x}$ and classification label $\boldsymbol{y}$, our explanation assigns a relevance (importance) value for each input feature, creating a saliency map for $(\boldsymbol{x}, \boldsymbol{y})$. The method we develop for explaining the classification results and constructing the corresponding saliency map is based on the technique introduced by Zintgraf et al. (2017), also referred to as prediction difference analysis (PDA). PDA is a robust and probabilistically sound method that can provide high-quality saliency maps for image datasets of different kinds (benchmarks, medical data, etc). Its run-time overhead was modest for the network architectures we used. The relevance of each feature is quantified via the counterfactual hypothesis designating how much the prediction would have changed had this particular feature not been involved in training and prediction (Robnik-Sikonja & Kononenko, 2008). As such, the relevance of a feature $\boldsymbol{x_i}$[2] is proportionate to

---

[1] Here we suppress the dependence on the inputs in $p(\boldsymbol{\theta}|\mathcal{D}_{1:t})$ and $p(\mathcal{D}_t|\boldsymbol{\theta})$ to lighten notation.

[2] In a slight abuse of notation, we use $\boldsymbol{i}$ here to refer to the index of an image pixel (or a square of adjacent pixels); which is a simplification of the row and column indices, $\boldsymbol{j}, \boldsymbol{k}$, used also to refer to a particular index of an image feature later on.

the difference between the predictions $p(\boldsymbol{y}|\boldsymbol{x}) = \int_{\boldsymbol{\theta}} p(\boldsymbol{y}|\boldsymbol{\theta}, \boldsymbol{x})p(\boldsymbol{\theta})$ and $p(\boldsymbol{y}|\boldsymbol{x}_{\backslash i})$, where $\boldsymbol{x}_{\backslash i}$ refers to the set of all features except $\boldsymbol{x}_i$. Since it is prohibitively intractable to start the whole training procedure of the classifier over again to compute $p(\boldsymbol{y}|\boldsymbol{x}_{\backslash i})$, for every $\boldsymbol{x}_{\backslash i}$, the PDA algorithm mimics the absence of $\boldsymbol{x}_{\backslash i}$ in training by marginalising the feature as follows:

$$p(\boldsymbol{y}|\boldsymbol{x}_{\backslash i}) = \sum_{\boldsymbol{x}_i} p(\boldsymbol{x}_i|\boldsymbol{x}_{\backslash i})p(\boldsymbol{y}|\boldsymbol{x}_{\backslash i}, \boldsymbol{x}_i). \tag{3}$$

The expression in equation 3 can be obtained by simply applying the Bayes product and sum rules to $p(\boldsymbol{y}|\boldsymbol{x}_{\backslash i}) = \sum_{\boldsymbol{x}_i} p(\boldsymbol{y}, \boldsymbol{x}_i|\boldsymbol{x}_{\backslash i})$. To compute $p(\boldsymbol{x}_i|\boldsymbol{x}_{\backslash i})$, we use an assumption that is usually valid for image data. The value of an image pixel depends very strongly on a small neighbourhood around it (Zintgraf et al., 2017), and this is true regardless of the pixel's position in the image. We therefore approximate $p(\boldsymbol{x}_i|\boldsymbol{x}_{\backslash i})$ with $p(\boldsymbol{x}_i|\hat{\boldsymbol{x}}_{\backslash i})$, where $\hat{\boldsymbol{x}}_i$ is a patch containing $\boldsymbol{x}_i$. Then, from equation 3:

$$p(\boldsymbol{y}|\boldsymbol{x}_{\backslash i}) \approx \sum_{\boldsymbol{x}_i} p(\boldsymbol{x}_i|\hat{\boldsymbol{x}}_{\backslash i})p(\boldsymbol{y}|\boldsymbol{x}_{\backslash i}, \boldsymbol{x}_i). \tag{4}$$

Define the odds of a prediction $(\boldsymbol{y}|\boldsymbol{x})$ as $\mathrm{odds}(\boldsymbol{y}|\boldsymbol{x}) = p(\boldsymbol{y}|\boldsymbol{x})/(1 - p(\boldsymbol{y}|\boldsymbol{x}))$. To get a sense of its effect: odds of $(\boldsymbol{y}|\boldsymbol{x})$ has a value of 0 when $p(\boldsymbol{y}|\boldsymbol{x}) = 0$ and tends to $\infty$ when $p(\boldsymbol{y}|\boldsymbol{x}) = 1$. We use Laplace correction $p \leftarrow (pN + 1)/(N + |\boldsymbol{Y}|)$, where $N$ is the number of training instances of the respective task, and $|\boldsymbol{Y}|$ is the number of classes in the task. Finally, we can compute the relevance of $\boldsymbol{x}_{\backslash i}$ by assessing the difference between $p(\boldsymbol{y}|\boldsymbol{x})$ and $p(\boldsymbol{y}|\boldsymbol{x}_{\backslash i})$, using a notion referred to as weight of evidence (WE, Robnik-Sikonja & Kononenko, 2008):

$$\mathrm{WE}_i(\boldsymbol{y}|\boldsymbol{x}) = \log_2\left(\mathrm{odds}(\boldsymbol{y}|\boldsymbol{x})\right) - \log_2\left(\mathrm{odds}(\boldsymbol{y}|\boldsymbol{x}_{\backslash i})\right). \tag{5}$$

The magnitude and sign of $\mathrm{WE}_i$ denote how significant the contribution of $\boldsymbol{x}_i$ is to this prediction. A small $\mathrm{WE}_i$ means that $\boldsymbol{x}_i$ was not influential in the prediction procedure, and vice versa. A positive (negative) valued $\mathrm{WE}_i$ indicates that $\boldsymbol{x}_i$ provides positive evidence for (against) the prediction of class $\boldsymbol{y}$. Assume that an input image (data point) $\boldsymbol{x}$ has $\boldsymbol{r}$ rows and $\boldsymbol{c}$ columns. Since PDA produces a relevance value for each feature, $\mathrm{WE}_i(\boldsymbol{y}|\boldsymbol{x})$ is also a matrix of dimensions $\boldsymbol{r} \times \boldsymbol{c}$.

## 4 Learning from Past Tasks via Saliency Based Attention

Explanations of predictions can enhance understanding and trust. Additionally, we show that PDA explanations can be used to improve performance on subsequent tasks. The feature relevance values obtained by PDA are used to build an attention mechanism, used when training on the next task to focus on important parts of the input. Tasks presented to the learner in the continual learning setting evolve over time. Thus, we assume that adjacent tasks are similar.

Our attention strategy is described as follows: After having finished the training procedure of a task, we gain further insights on the corresponding classification predictions on a test set via the explanations provided by PDA. The outcome of PDA is a saliency map for every test image. We summarise these results by computing their average to obtain a saliency map representing averaged relevance values of the input features for the finished task.[3] Since the upcoming task is similar, we utilise the information already at our disposal by developing an attention mechanism based on the computed relevance values from the immediately preceding task, described as follows.

For input images of the upcoming task $\boldsymbol{x} \in \mathbb{R}^{\boldsymbol{r} \times \boldsymbol{c}}$, the averaged weight of evidence matrix is referred to as $\mathrm{WE}_i(\boldsymbol{x}) \in \mathbb{R}^{\boldsymbol{r} \times \boldsymbol{c}}$. An attention mask $\boldsymbol{M} \in \mathbb{R}^{\boldsymbol{r} \times \boldsymbol{c}}$ is inferred as a function of the WE values. For a position in the image $\boldsymbol{j}, \boldsymbol{k}$, with row and column indices $\boldsymbol{j}$ and $\boldsymbol{k}$, the mask value $\boldsymbol{M}_{\boldsymbol{j},\boldsymbol{k}}$ can be computed as follows, where $\boldsymbol{z}$ denotes an offset used to smooth out the mask computation.

$$\boldsymbol{M}_{\boldsymbol{j},\boldsymbol{k}} = \frac{\sum_{u=-z}^{u=z} \sum_{v=-z}^{v=z} \boldsymbol{x}_{\boldsymbol{j}+u,\boldsymbol{k}+v} \, \mathrm{WE}_{\boldsymbol{j}+u,\boldsymbol{k}+v}}{(2\,\boldsymbol{z}+1)^2}. \tag{6}$$

For each pixel in the image, the mask is composed of an average of the relevance values of a square around the pixel, with side $2\boldsymbol{z}$. For pixel positions at the border, e.g. $\boldsymbol{j}$ or $\boldsymbol{k} = 1$, the denominator is changed always to be the number of elements used. The mask is then used with the training data values in the training procedure of the subsequent task to help focus on important locations.

---

[3]This averaging procedure is effective in our current experiments, but we recognize that for some datasets, e.g. in settings where many images are rotated, a more sophisticated summarization approach may be desirable. For instance, a clever normalization strategy can help enhance the invariance properties of the attention mechanism. We leave this as a direction to explore in future work.

## 5 EVALUATION OF SALIENCY MAPS

One contribution of ICL is the ability to assess how much catastrophic forgetting is taking place via evaluating the difference between the explanations of a specific task delivered when the task at hand is the most recent, and the explanation of the same task after having learnt other (more recent) tasks. This requires a concrete way to evaluate saliency maps, which is the main concern of this section.

We first describe a related earlier method by Dabkowski & Gal (2017). Their approach is based on a notion of saliency referred to as the Smallest Sufficient Region (SSR), defined as the smallest region in the image that, when introduced as input to the classifier on its own without the rest of the image, leads to a considerably confident classification (Fong & Vedaldi, 2017). According to this notion, a saliency map is better if it is capable of producing a relevant region in the image that is: i) small, and that ii) leads to a confident prediction (when the classifier acts solely on the relevant region, rather than the whole image). Since the metric suggested in (Dabkowski & Gal, 2017) needs a fixed-shape input to the image classifier, a rectangle representing the superset of the relevant region in the image is selected as the region of interest. This rectangle is chosen as the smallest rectangle containing the entire salient region. Denote by $\mathbf{a}$ the area of this relevant rectangle (the superset of the region containing all the salient pixels), and denote by $\tilde{\mathbf{a}} = \max(\mathbf{a}, 0.05)$ a threshold for the area $\mathbf{a}$ used to prevent instabilities resulting from very small area values. Also let $p$ refer to the prediction probability returned by the classifier for the specified label. The SSR saliency metric $\mathbf{s}(\mathbf{a}, p)$ can then be expressed as follows:

$$\mathbf{s}(\mathbf{a}, p) = \log(\tilde{\mathbf{a}}) - \log(p). \qquad (7)$$

For the saliency metric defined in equation 7, the lower the value of $\mathbf{s}(\mathbf{a}, p)$ the better. A low value of $\mathbf{s}(\mathbf{a}, p)$ signifies a small area of the relevant rectangle $\mathbf{a}$ as well as a confident prediction $p$ by the classifier. It remains tricky how to identify the salient area, which is bounded by $\mathbf{a}$, what kind of threshold shall be used to indicate a salient vs. non-salient values, etc. In addition, this area has to be one connected chunk, potentially wasting the opportunity to identify salient, but possibly non-connected, areas like the two eyes of an animal, etc. Furthermore, choosing the entire salient region as a connected region $\mathbf{a}$ can lead to including several non-salient areas therein.

Here we propose a new saliency map metric that is less restricted by the issues above. Our saliency metric has more freedom in indicating the salient pixels based on merit regardless of their location and their vicinity to other salient pixels. We believe that the vicinity among salient pixels should in fact be used as a part of the metric, i.e. a deciding factor in evaluating saliency maps, rather than be forced to identify the salient regions in the first place. We therefore propose basing the evaluation on three aspects, number of salient pixels (regardless of their locations), average distance among the salient pixels, and their impact on the classification prediction. For the former two aspects (number and average distance among the salient pixels) the lower the values the better the quality of the resulting saliency. The opposite is true (negative sign) for the impact on the prediction since salient pixels of a better map have a higher impact. One of the problems that used to prohibit a free choice of the salient pixels regardless of their adjacency to each other, is that almost all image classifiers need an image of a fixed size to work on. We have a built-in solution to that since PDA provides the facility to marginalise pixels, and that is what we follow in order to assess the impact of the salient pixels on prediction; we marginalise over the salient pixels to analyse their impact.

Our saliency metric, which we refer to as, the flexible saliency metric (FSM) is defined as:

$$\mathrm{fsm}(p) = \log(\boldsymbol{d}_{\mathrm{sal}}) + \log(\boldsymbol{m}) - \log(p), \qquad (8)$$

where $\boldsymbol{d}_{\mathrm{sal}}$ refers to the average spatial distance among the salient pixels, $\boldsymbol{m}$ refers to the number of the salient pixels and $p$ refers to the classification probability of the specified label.

## 6 EXPERIMENTS

We provide quantitative and qualitative evaluations of the proposed ICL. We perform experiments on three datasets: MNIST (LeCun et al., 1998), notMNIST (Butalov, 2011) and Fashion-MNIST (Xiao et al., 2017). Our experiments mainly aim at evaluating the following: (i) performance of the introduced ICL framework depicted by the classification accuracy; (ii) quality of the explanations provided by ICL in the form of saliency maps. This is quantitatively measured by the proposed metric. The extent to which catastrophic forgetting can be mitigated when deploying ICL or VCL can be

inspected via evaluating the respective classification accuracy and saliency maps; and (iii) saliency examples qualitatively showing that the explanations do not suffer from catastrophic forgetting.

In ICL, we extract explanations after learning each task and use them as a precursor to the attention mechanism for the following task. After developing the mask needed for the attention mechanism, VCL is used for the task learning. In Section 6.1, we compare the classification results obtained by ICL to three different (non-interpretable) versions of the VCL algorithm (Nguyen et al., 2018) and to the elastic weight consolidation (EWC) algorithm (Kirkpatrick et al., 2017). As mentioned earlier, one of the strengths of ICL is that it can be applied to many continual learning frameworks. We base our original ICL version on VCL due to advantages of the latter in terms of flexibility, being effective at reducing catastrophic forgetting, etc. When evaluating the algorithm in terms of the introduced interpretability metric in Section 6.2, we compare the standard VCL and EWC to interpretable (ICL) versions of both. For all experiments, statistics reported are averages of ten repetitions. Statistical significance of the the average accuracy and FSM results obtained after completing the last two tasks from each dataset are displayed in the tables in Appendix A. We also display the results of a comparison with a fixed attention map, consisting of concentric circles, as a baseline in Appendix A. Regarding PDA, 10 samples are used to estimate $p(\boldsymbol{y}|\boldsymbol{x}_{\setminus i})$ (equation 4). The size of the square of pixels marginalised at once, $\boldsymbol{x_i}$, is $10 \times 10$. The size of the surrounding square $\hat{x}_{\boldsymbol{i}}$ used to approximate $p(\boldsymbol{x_i}|\boldsymbol{x}_{\setminus i})$, is $16 \times 16$ pixels. The only exception to the latter is the Permuted MNIST experiment where a larger $\hat{x}_{\boldsymbol{i}}$ of size $20 \times 20$ is needed to mitigate the permutation impact.

## 6.1 CLASSIFICATION ACCURACY

We compare the all-important classification accuracy of ICL to state-of-the-art continual learning frameworks. We aim at assessing the impact of adopting the introduced attention mechanism, which is based on the classification explanations provided by the saliency maps, on top of the seminal VCL algorithm. We consider four continual learning experiments, based on the MNIST, notMNIST and Fashion-MNIST datasets. The introduced ICL is compared to EWC in addition to the following non-interpretable versions of VCL: VCL with no coreset, VCL with a random coreset consisting of 200 points[4], VCL with a 200-point coreset assembled by the K-center method (Nguyen et al., 2018). All the reported classification accuracy values reflect an average of the classification accuracy over all tasks the learner has trained on so far. More specifically, assume that the continual learner has just finished training on a task $t$, then the reported classification accuracy at time $t$ is the average accuracy value obtained from testing on equally sized sets each belonging to one of the tasks 1, 2, ..., $t$. ICL achieves state-of-the-art classification accuracy in three out of the four experiments, and it achieves joint highest accuracy in the fourth (Permuted MNIST) with VCL with a random coreset.

**Permuted MNIST** Using MNIST, Permuted MNIST is a standard continual learning benchmark (Goodfellow et al., 2014; Kirkpatrick et al., 2017; Zenke et al., 2017). For each task $t$, the corresponding dataset is formed by performing a fixed random permutation process on labeled MNIST images. This random permutation is unique per task, i.e. it differs for each task. For the hyperparameter $\lambda$ of EWC, which controls the overall contribution from previous data, we experiment with two values, $\lambda = 1$ and $\lambda = 100$. The latter has previously produced the best EWC classification results (Nguyen et al., 2018). In this experiment, fully connected single-head networks with two hidden layers are used. There are 100 hidden units in each layer, with ReLU activations. Results of the accumulated classification accuracy, averaged over tasks, on a test set are displayed in Figure 1. After 10 tasks, ICL and VCL using a random coreset achieve the highest classification accuracy.

**Split MNIST** In this MNIST based experiment, five binary classification tasks are processed in the following sequence: 0/1, 2/3, 4/5, 6/7, and 8/9 (Zenke et al., 2017). The architecture used consists of fully connected multi-head networks with two hidden layers, each consisting of 256 hidden units with ReLU activations. As can be seen in Figure 1, ICL achieves the highest classification accuracy.

**Split notMNIST** This is similar to the last one, but the dataset used (notMNIST) is larger and a bit more challenging. It contains 400,000 training images, and the classes are 10 characters, from A to J. Each image consists of one character, and there are different font styles. The five binary classification tasks are: A/F, B/G, C/H, D/I, and E/J. The networks used here contain four hidden

---

[4]Whenever there is no validation process performed to indicate the hyperparameter values of competitors or characteristics of neural network architectures, this is done for the sake of comparing on common ground with the best settings, as specified in the respective papers.

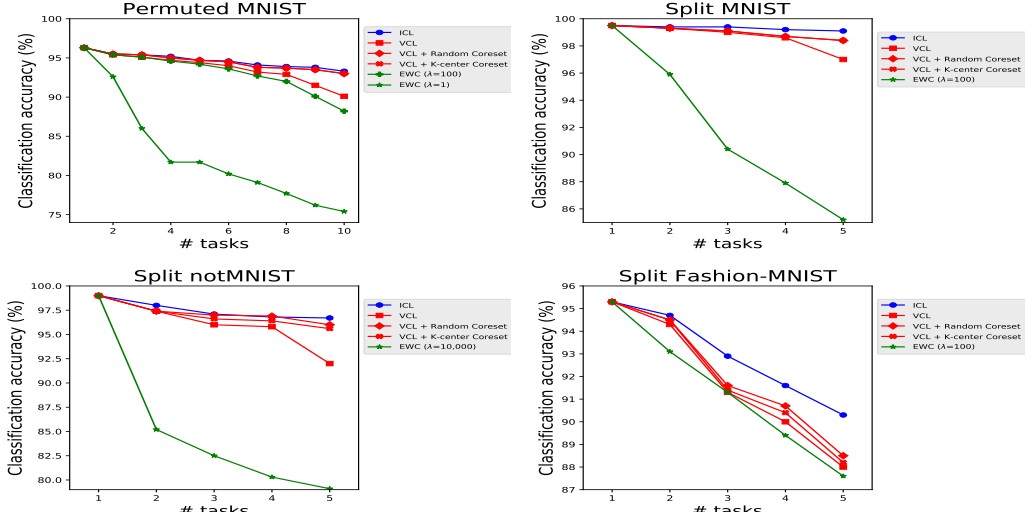

Figure 1: Average test classification accuracy vs. the number of observed tasks in the Permuted MNIST, Split MNIST, Split notMNIST, and Split Fashion-MNIST experiments.

layers, each containing 150 hidden units with ReLU activations. ICL achieves an improvement in classification accuracy over competitors (Figure 1). VCL with a random coreset achieves 96.1% accuracy, whereas ICL accomplishes an average accuracy of 96.5%.

**Split Fashion-MNIST** This is another similar but more challenging dataset than MNIST. Fashion-MNIST is a dataset whose size is the same as MNIST but it is based on different 10 classes. The five binary classification tasks here are: T-shirt/Trouser, Pullover/Dress, Coat/Sandals, Shirt/Sneaker, and Bag/Ankle boots. The architecture used is the same as in Split notMNIST. ICL achieves 90.3% average accuracy after 5 tasks, while VCL with a random coreset attains 88.5% (Figure 1).

## 6.2 EVALUATION OF CLASSIFICATION EXPLANATIONS

Based on the saliency metric proposed in equation 8, and referred to as FSM, we evaluate the quality of the explanations resulting from ICL. We compare the non-interpretable versions of VCL and EWC to interpretable (ICL) versions of both. The lower the FSM value the better.

In Figures 2, 3 and 4, results of the FSM saliency map metric are displayed for the Split MNIST, Split notMNIST and Split Fashion-MNIST experiments, respectively. The averaged results of the five tasks in each of the three experiments (as well as most of the individual tasks) show that ICL, when used with VCL, leads to a better (lower) FSM metric value. This signifies two major empirical findings. First, ICL yields more interpretable learning procedures, depicted by more interpretable maps. Second, since the averaged results take into consideration less recent tasks, this means that ICL remains interpretable, even after facing new learning tasks, which -along with the classification results in Section 6.1- means that ICL, when used with VCL, is less prone to catastrophic forgetting.

## 6.3 EXAMPLES OF EXPLANATIONS

We display some examples demonstrating the efficacy of ICL in mitigating catastrophic forgetting via comparing the explanations provided for a classification decision right after learning the corresponding task, with the same explanation after eventually performing the training of some other tasks. The latter represents the explanation potentially prone to catastrophic forgetting. In each of the examples we provide, we begin with showing the original test image, followed by the explanation provided by the task right after it has been learnt. Finally, the column at the far right provides the explanation of the same task that is prone to catastrophic forgetting, i.e. the one provided by the learner after learning other tasks. The fact that most of the explanations remain similar is a qualitative demonstration of the efficacy of ICL, when applied to VCL, in mitigating catastrophic forgetting. Five examples are shown in Figure 5. All explanations provided are for test images that have been predicted correctly by the learner in both cases (both time steps).

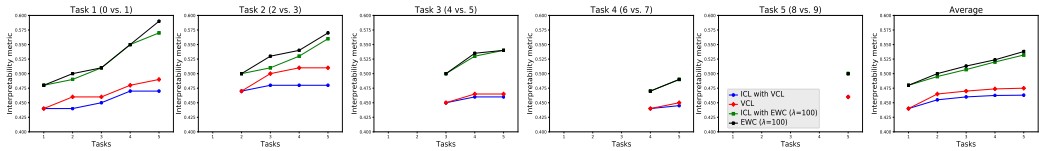

Figure 2: Values of the proposed saliency metric FSM (equation 8) on the 5 tasks in Split MNIST. Lower is better. The last (bottom right) column displays the average accuracy over all tasks.

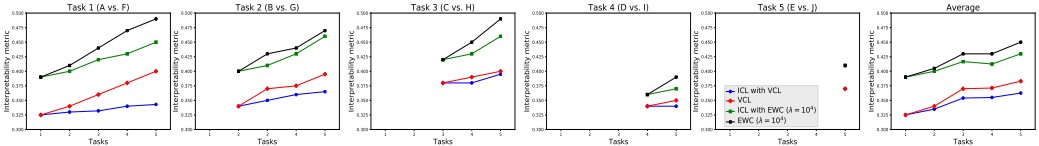

Figure 3: Values of the proposed saliency metric FSM (equation 8) on the 5 tasks in Split notMNIST. Lower is better. The last (bottom right) column displays the average accuracy over all tasks.

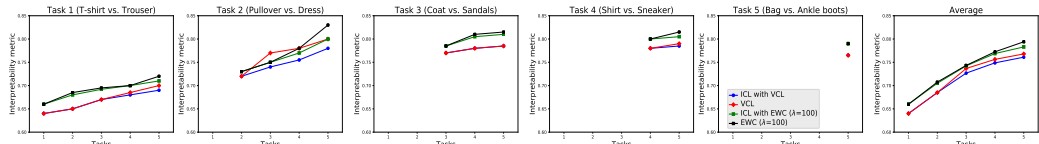

Figure 4: Values of the proposed saliency metric FSM on the 5 tasks in Split Fashion-MNIST. Lower is better. The last (bottom right) column displays the average accuracy over all tasks.

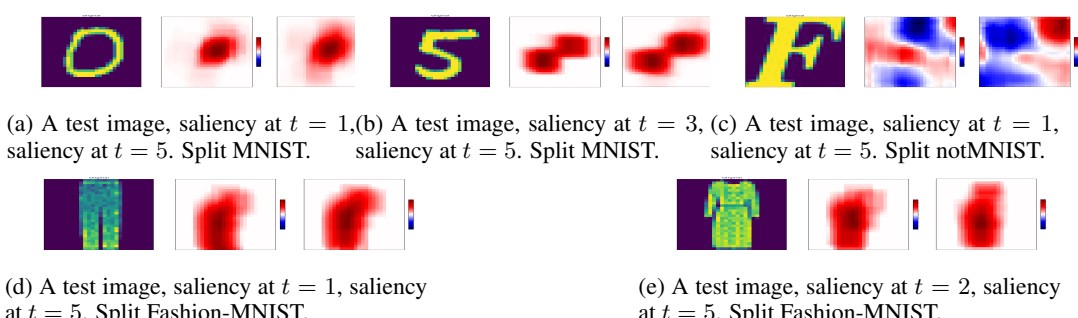

(a) A test image, saliency at $t = 1$,(b) A test image, saliency at $t = 3$, (c) A test image, saliency at $t = 1$, saliency at $t = 5$. Split MNIST. saliency at $t = 5$. Split MNIST. saliency at $t = 5$. Split notMNIST.

(d) A test image, saliency at $t = 1$, saliency at $t = 5$. Split Fashion-MNIST.

(e) A test image, saliency at $t = 2$, saliency at $t = 5$. Split Fashion-MNIST.

Figure 5: Explanations (in the form of saliency maps) of the classification prediction on a test image from a specific task in a particular experiment. Evidence for (against) the predicted class is shown in red (blue). The middle map represents the explanation at the time training has just been finished for this task, whereas the map at the right side of each subfigure is the explanation at $t = 5$.

## 7 CONCLUSION

We introduced a continual learning framework incorporating interpretability, where saliency based explanations of previously learnt tasks are used to enhance the attention of the learner during future tasks. This framework demonstrates that interpretability is not only useful for increasing the understanding of the obtained results, but can also improve the performance of a sequential learning procedure. The proposed framework is flexible and can enhance both the interpretability and performance of continual learning methods, especially in terms of mitigating catastrophic forgetting. We proposed a new metric for saliency maps. We believe that adopting a Bayesian attention mechanism could be a fruitful direction for future work, especially when integrated with fully Bayesian variational continual learning.

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

APPENDIX A

We first display the results of the highest performing competing frameworks in terms of the average accuracy values (Figure 1) obtained after completing the last two tasks from each of the four experiments in Table 1. In addition, we compare to a predefined, fixed attention map consisting of concentric circles where the centre is the image centre. Radius values in use have been indicated by cross-validation. A bold entry in Table 1 denotes that the classification accuracy of an algorithm is significantly higher than its competitors. Significant results are identified using a paired t-test with p = 0.05. Average classification accuracy resulting from ICL is significantly higher than its competitors in 6 out of the 8 comparison cases. Also, deterioration of the accuracy values due to using the fixed attention map empirically demonstrates the significance of the learning procedure adopted to estimate the ICL attention maps.

Table 1: Average test classification accuracy of 4 and 5 tasks for the Split MNIST, Split notMNIST and Split Fashion-MNIST experiments, and for 9 and 10 tasks of Permuted MNIST. A bold entry in Table 1 denotes that the classification accuracy of an algorithm is significantly higher than its competitors. Significant results are identified using a paired t-test with p = 0.05. Average classification accuracy resulting from ICL is significantly higher than its competitors in 6 out of the 8 cases (3 out of 4 after the completion of all the tasks required per experiment).

| Classification accuracy | ICL | VCL | VCL + Random Coreset | VCL + K-center Coreset | VCL + fixed attention |
|---|---|---|---|---|---|
| Permuted MNIST (task 9) | **93.8**% | 91.5% | 93.5% | 93.5% | 92.9% |
| Permuted MNIST (task 10) | 93.3% | 90.1% | 93% | 93% | 92.2% |
| Split MNIST (task 4) | **99.2**% | 98.6% | 98.7% | 98.7% | 97.1% |
| Split MNIST (task 5) | **99.1**% | 97% | 98.4% | 98.4% | 96.8% |
| Split notMNIST (task 4) | 96.8% | 95.8% | 96.9% | 96.4% | 95.2% |
| Split notMNIST (task 5) | **96.7**% | 92% | 96% | 95.6% | 95% |
| Split Fashion-MNIST (task 4) | **91.6**% | 90% | 90.7% | 90.4% | 87.7% |
| Split Fashion-MNIST (task 5) | **90.3**% | 88% | 88.5% | 88.2% | 86.3% |

We then shed light again on the results of the FSM interpretability metric -Figures 2, 3 and 4- in Table 2 to display the statistical significance unequivocally. Recall that lower FSM values are better. A result in bold refers to an FSM value that is significantly lower (better) than the competitors. FSM values resulting from using ICL with VCL are significantly better in all of the 6 cases.

Table 2: Values of the proposed saliency metric FSM for the Split MNIST, Split notMNIST and Split Fashion-MNIST experiments. A result in bold refers to an FSM value that is significantly lower (better) than the other competitors. FSM values resulting from using ICL with VCL are significantly better in all of the 6 cases.

| Interpretability metric | ICL with VCL | VCL | ICL with EWC ($\lambda$=100) | EWC ($\lambda$=100) |
|---|---|---|---|---|
| Split MNIST (task 4) | **0.4625** | 0.4738 | 0.52 | 0.5238 |
| Split MNIST (task 5) | **0.463** | 0.475 | 0.532 | 0.538 |
| | ICL with VCL | VCL | ICL with EWC ($\lambda = 10^4$) | EWC ($\lambda = 10^4$) |
| Split notMNIST (task 4) | **0.355** | 0.3713 | 0.4125 | 0.43 |
| Split notMNIST (task 5) | **0.3626** | 0.383 | 0.43 | 0.45 |
| | ICL with VCL | VCL | ICL with EWC ($\lambda$=100) | EWC ($\lambda$=100) |
| Split Fashion-MNIST (task 4) | **0.7488** | 0.7562 | 0.7688 | 0.7725 |
| Split Fashion-MNIST (task 5) | **0.761** | 0.768 | 0.783 | 0.794 |

