# OpenReview forum: "Interpretable Continual Learning"
_ICLR.cc/2019/Conference_

### Official Review · AnonReviewer1 · 2018-11-02
**Reasonable approach with good results; incremental novelty**

**Rating:** 6
**Confidence:** 3

**Review:**

This paper proposes an extension to the continual learning framework using existing variational continual learning (VCL) as the base method. In particular, it proposes to use the weight of evidence (WE) (from Zintgraf et al 2017) for each task. Firstly, this WE can be used to visualize the learned model (as used in Zintgraf et. al. 2017). The novelty of this paper is:
1. to use this WE from the current task to generate a silence map (by smoothing the WE) for the next task.  This is interpreted the learned the learned attention region. Such an approach is named Interpretable COntinual Learning (ICL)
2. The paper proposes a metric for the saliency map naming FSM which is an extension of existing metric SSR. The extension is to take pixel count to compute the area instead of using rectangular region area, as well as taking the distance between pixels into account. This metric can be used to evaluate the level of catastrophic forgetting.

Pro:
In general, the idea is very intuitive and make sense.  The paper also demonstrates superior performance with the proposed method on continual learning on all classic tasks comparing with VCL and EWC.
The presentation is very easy to follow.
It seems like a valid and flexible extension that can be used in other continual learning frameworks.

Cons:
The theoretical contribution is very limited. The work is rather incremental from current state-of-the-art methods.
There should be a better discussion of related work on the topic. The paper currently only mentions the most related work for the proposed method,  using the whole section 2 to describe VCL and use section 3 to describe FSM and half of section 5 to describe SSR. A general overview of related work in these directions are needed.

Other:
1. The paper should also consider more recently proposed evaluation metrics such as discussed in https://arxiv.org/pdf/1805.09733.pdf
2. The author should try to avoid using yellow color in plots.

---

> ### Author Response · Authors · 2018-11-10
> **Response to AnonReviewer1**
>
> We thank the reviewer for their time and welcome feedback, which we are incorporating into the revised version.
>
> R: - Theoretical contribution
> A:
> -- Just a clarification regarding the first contribution: WE from the current task is not used to generate a saliency map for the next task; it is instead used to instruct the learner of the next task about which input areas are more important than others via the attention mechanism. This becomes a part of the future learning procedure, not just a post hoc visualisation method as in the original WE . As such, the first contribution is not solely about generating new visualisations; it is more about using the learned saliency maps from the past to attend in future learners. We therefore believe that the potential of this first contribution as a conceptual framework via which a learner can understand, attend, and then enhance its attention for future tasks, is not small.
> -- Importantly, we are the first to combine interpretability with continual learning and show that interpretability can help continual learning. It is a significant step to bring these two communities together.
> -- It is worth noting that VCL has achieved very good results on most of these benchmarks, so it is very hard to outperform it with a large margin. Nevertheless, although the experimental increases might not seem dramatic, they are statistically significant (we have added the statistical significance results to the appendix in the revised version).
>
>
> R: - Related work
> A:
> -- Thank you. In addition to the locations specified by the reviewer, the first two paragraphs of Section 1 (especially the second paragraph) discuss several other related works. We have added more to the revised version.
>
>
> R: - https://arxiv.org/pdf/1805.09733.pdf
> A:
> -- Thank you. We have cited the paper in the revised version, and plan to take it into further consideration in future work.
>
> R: - Yellow color in plots
> A:
> -- We have changed the yellow colour in Figures 2, 3 and 4 to black. Yellow is now no longer used in any plots.

---

### Official Review · AnonReviewer3 · 2018-11-04
**A paper with a relevant and interesting contribution that lacks clarity and motivation**

**Rating:** 5
**Confidence:** 3

**Review:**

Summary:
In this paper, the authors propose a framework for continual learning based on explanations for performed classifications of previously learned tasks. In this framework, an average saliency map is computed for all images in the test set of a previous task to identify image regions, which are important for that task. When learning the next task, this average saliency map is used in an attention mechanism to help learning the new task and to prevent catastrophic forgetting of previously learned tasks. Furthermore, the authors propose a new metric for the goodness of a saliency map by taking into account the number of pixels in the map, the average distance between pixels in the map, as well as the prediction probability given only the salient pixels.
The authors report that their approach achieves the best average classification accuracy for 3 out of 4 benchmark datasets compared to other state-of-the-art approaches.

Relevance:
This work is relevant to researchers in the field of continual/life-long learning, since it proposes a framework, which should be possible to integrate into different approaches in this field.


Significance:
The proposed work is significant, since it explores a new direction of using learner generated, interpretable explanations of the currently learned task as help for learning new tasks. Furthermore, it proposes a new metric for the goodness of saliency maps.


Soundness:
In general, the proposed approach of using the average saliency map as attention mask for learning appears to be reasonable. However, the following implicit assumptions/limitations of the approach should be made more clear:
	- important features for the new task should be in similar locations as important features of the old task (for example, one would expect that the proposed approach would negatively affect learning the new task if the important features of the old task were all located in the bottom of the image, while all important features for the new task are in the top)
	- the locations for important features should be comparatively stable (for example, one would expect the average saliency map to become fairly meaningless if important features, such as the face of a dog, can appear anywhere in the image. Therefore, an interesting baseline for the evaluation of the ICL approach would be a predefined, fixed attention map consisting of concentric circles with the image center as their center, to show that the proposed approach does more than just deemphasizing the corners of the image)

Furthermore, the authors appear to imply that increased FSM values for an old task after training on a new task indicate catastrophic forgetting. While this is a reasonable assumption, it does not necessarily seem to be the case that a larger, more disconnected saliency map indicates worse classification performance. Comparatively small changes in FSM may not affect the classification performance at all, while larger changes may not necessarily lead to worse classifications either. For example, by increasing the amount or size of image regions to be considered, the classifier may accidentally become more robust on an old task. Therefore, it may be a good idea for the authors to analyze the correlation between FSM changes and accuracy changes.

Evaluation:
The evaluation of the proposed approach on the four used datasets appears to be reasonable and well done. However, given that the achieved performance gains over the state-of-the-art are fairly small, it would be good to assess if the obtained improvements are statistically significant.
Furthermore, it may be informative to show the saliency maps in Figure 5 not only for cases in which the learner classified the image correctly in both time steps, but also cases in which the learner classified the image correctly the first time and incorrectly the second time. Additionally, the previously mentioned evaluation steps, i.e., using a fixed attention map as baseline for the evaluation and evaluating the correlation between FSM and accuracy may be informative to illustrate the advantages of the proposed approach.

Clarity:
The paper is clearly written and easy to follow. One minor issue is that the first sentence of the third paragraph in Section 4 is not a full sentence and therefore difficult to understand.
Furthermore, on page 6, it is stated that the surrounding square $\hat{x}_i$ is 15 x 15 pixels, while the size of the square $x_i$ is 10 x 10. This appears strange, since it would mean that $x_i$ cannot be in the center of $\hat{x}_i$.

---

> ### Author Response · Authors · 2018-11-10
> **Response to AnonReviewer3**
>
> We thank the reviewer for their time and welcome feedback, which we are incorporating into the revised version.
>
> R: - "important features for the new task should be in similar locations ..."
> - "the locations for important features should be comparatively stable ..."
> A:
> -- Continual learning typically assumes a degree of similarity among the tasks. If tasks are completely different from each other, then most continual learning frameworks will somehow struggle. For example, the standard Split MNIST benchmark is in line with this “locations of important features” assumption. Having said that, we acknowledge that more agility to, at least, discover that early on would be beneficial. More importantly, a normalization strategy on top of our attention map would help enhance its invariance properties, potentially leading to a more robust treatment of the locations of important features. In page 4 in the revised version (footnote 3), we have clarified this and notified its potential for future work.
> -- Thank you for the suggestion regarding the fixed attention map. We tried an experiment using the fixed attention map as a baseline, and as expected it performs significantly worse than ours. We have added that to the revised version (see p.6 and Appendix A).
>
>
> R: - FSM vs. Classification performance
> A:
> -- It is true that evaluating the FSM is not necessarily the same as the classification results, which is precisely the reason why we show both in our results. As specified in page 2, “Here we propose a new measure ...” - our point in this regard is to propose another (different) manner via which catastrophic forgetting can be estimated, which is not the same as the classification accuracy. The goal is that (as we know and agree they are two different measures that might agree or disagree in their judgments on catastrophic forgetting) both can be used to inspect the degree of catastrophic forgetting. We have further clarified that in Section 6.2 in the experiments by stressing that the obtained FSM results “along with the classification results” denote the significance of the whole framework in addressing catastrophic forgetting.
> It is definitely a good idea to analyse the correlation between changes in classification accuracy and in FSM values, thank you. We will rigorously investigate this in future work.
>
>
> R: - Statistical significance
> A:
> -- Thank you. We have added the statistical significance results to the revised version. Since we were concerned that adding this information to the plots would make them harder to read,  statistical significance of the the average accuracy and FSM results obtained after completing the last two tasks from each dataset, i.e. the corresponding values of the last two tasks of all the plots in Figures 1, 2, 3 and 4, are now displayed in the tables in Appendix A.
> Checking cases where the learner incorrectly classifies the image in the second time step is sound and will be inspected in future work.
>
>
> R: - Clarity
> A:
> -- We have fixed the typos in the revised version, thank you: i) The first sentence of the third paragraph in Section 4 now reads: “For input images of ..., the averaged weight of evidence matrix  is referred to as $\text{WE}_{\bm{i}}(\bm{x}) \in \RR^{\bm{r} \times \bm{c}}$.”  ii) In page 6: “The size of the surrounding square … is 16 $\times$ 16 pixels.

---

### Official Review · AnonReviewer2 · 2018-11-15
**Interpretable Continual Learning**

**Rating:** 4
**Confidence:** 4

**Review:**

Authors propose an incremental continual learning framework which is based on saliency maps on the learned tasks(i.e., explanations) with the ultimate goal of learning new tasks, while avoiding catastrophic forgetting. To this end, authors employ an attention mechanism based on average saliency masks computed on the predictions of the earlier task. In addition, a new metric, Flexible Saliency Metric (FSM) is proposed to evaluate the generated saliency maps. Authors employ three public, well-known datasets to evaluate the performance of their proposed framework.

The paper is well written and easy to follow. The methodology is sound and the results demonstrate that the proposed framework outperforms very recent conditional learning approaches. Nevertheless I have some major concerns with the methodology, proposed evaluation metric and experiments. Please find below my comments.

- Technical novelty is rather limited. Contribution is incremental with respect to previous works on CL, as they use the variational CL (VCL) framework of Nguyen at al, 2018 and the weight of evidence (WE), as used in Zintgraf et al., 2017, to compute the saliency maps. From these saliency maps, a mask is computed to focus the attention in subsequent tasks, by averaging the explanations. This, however, limits the applicability of the proposed framework to ‘similar’ images (as pointed out by the authors). Another limitation of this technique is that explanations on learned tasks should correspond, spatially, to meaningful/discriminative areas for new tasks. Otherwise, the use of explanations on this CL approach would not work.
- According to the authors, one of the limitations of known metrics to evaluate CL approaches is that ‘the area of the saliency regions should be all connected, wasting opportunity to identify salient but possibly non-connected areas, such as the two eyes of an animal’. Nevertheless, I do not see how this can be alleviated in the proposed FSM. The first term of eq (8), i.e., log(d_sal) will be large in the case of, for example, the two eyes of an animal, favouring again for connected saliency regions. How d_sal is computed? Is it a dense matrix between all pair of points?
- Being the FSM one of the main contributions of this work, experiments to assess its usability are insufficient. Authors should correlate the values obtained across the different CL frameworks with FSM to the actual performance in terms of precision/accuracy. Results demonstrate that the proposed ICL approach achieves the lowest values, in terms of FSM, but any interpretation can be done if it is not correlated with well established evaluation metrics.
- Furthermore, it would be interesting to see how this method performs in more complex datasets, such as ImageNet, where tasks within the continual learning process may differ a lot.
- I also feel the literature on CL is scarce and it does not motivate the choices of the manuscript. Authors should include a more detailed literature on this problem.

Minor comments

- In page 3, which is the difference between benchmarks and medical data, as datasets? Public medical data are also benchmarks.
- How the z value in eq (6) is found? An ablation study to see the impact on the final results would be interesting.
- In page 5, when describing the limitations of current methods for saliency map evaluation (‘It remains tricky how to identify,….,etc),what does etc mean? Please be more concise on the limitations.

---

### Meta-Review · Area_Chair1 · 2018-12-11
**Saliency maps utilized for continual learning, but concerns around novelty and performance improvements.**

**Confidence:** 4
**Recommendation:** Reject

**Metareview:**

The presented method proposes to use saliency maps as a component for an additional metric of forgetting in continual learning, and as a tool as additional information to improve learning on new tasks.

Pros:
+ R2 & R3: Clearly written and easy to follow.
+ R3: New metric to compare saliency masks
+ R3: Interesting idea to utilize previously learned saliency masks to augment learning new tasks.
+ R1: Performance improvements observed.

Cons:
- R1 & R2: Novelty is limited in the context of prior works in this field. Unanswered by authors.
- R2: Concerns around method's ability to use salient but disconnected components. Unanswered by authors.
- R2: Experiments needed on more realistic datasets, such as ImageNet. Unanswered by authors.
- R3: Performance gains are small.
- R1 & R2: Literature review is insufficient.

Reviewers are leaning reject, and R2's concerns have not been answered by the authors at all. Idea seems interesting, authors are encouraged to take into careful consideration the feedback from authors and continue their research.

---

> ### Author Response · Authors · 2018-12-21
> **Regarding R2**
>
> Thanks for the assessment. Regarding R2, this review arrived two weeks late. Thanks again.